# Morphological Changes in Astrocytes by Self-Oxidation of Dopamine to Polydopamine and Quantification of Dopamine through Multivariate Regression Analysis of Polydopamine Images

**DOI:** 10.3390/polym12112483

**Published:** 2020-10-26

**Authors:** Anik Karan, Elnaz Khezerlou, Farnaz Rezaei, Leon Iasemidis, Mark A. DeCoster

**Affiliations:** 1Biomedical Engineering Department, College of Engineering and Science, Louisiana Tech University, Ruston, LA 71270, USA; aka029@latech.edu (A.K.); ekh008@latech.edu (E.K.); leonidas@latech.edu (L.I.); 2Center for Biomedical Engineering and Rehabilitation Science, College of Engineering and Science, Louisiana Tech University, Ruston, LA 71270, USA; 3Computational Analysis and Modeling Program, College of Engineering and Science, Louisiana Tech University, Ruston, LA 71270, USA; fre004@latech.edu; 4Institute for Micromanufacturing, College of Engineering and Science, Louisiana Tech University, Ruston, LA 71270, USA

**Keywords:** astrocyte, polydopamine, dopamine, neurotransmitter, image analysis

## Abstract

Astrocytes, also known as astroglia, are important cells for the structural support of neurons as well as for biochemical balance in the central nervous system (CNS). In this study, the polymerization of dopamine (DA) to polydopamine (PDA) and its effect on astrocytes was investigated. The polymerization of DA, being directly proportional to the DA concentration, raises the prospect of detecting DA concentration from PDA optically using image-processing techniques. It was found here that DA, a naturally occurring neurotransmitter, significantly altered astrocyte cell number, morphology, and metabolism, compared to astrocytes in the absence of DA. Along with these effects on astrocytes, the polymerization of DA to PDA was tracked optically in the same cell culture wells. This polymerization process led to a unique methodology based on multivariate regression analysis that quantified the concentration of DA from optical images of astrocyte cell culture media. Therefore, this developed methodology, combined with conventional imaging equipment, could be used in place of high-end and expensive analytical chemistry instruments, such as spectrophotometry, mass spectrometry, and fluorescence techniques, for quantification of the concentration of DA after polymerization to PDA under in vitro and potentially in vivo conditions.

## 1. Introduction

Astrocytes are the glial cells in the central nervous system (CNS) which provide both structural and biochemical support to neurons. Astrocytes can take on many shapes, including stellated in appearance, and may outnumber neurons by over five-fold in the brain [1]. These support cells are involved in maintaining the overall chemical balance in the brain between neurons and other types of cell in a collectively linked systemic network. Other functions of astrocytes include biochemical support of brain microvascular endothelial cells in the blood brain barrier (BBB) [2], controlling cerebral blood flow [3], promoting immunomodulation [4], efficient transportation and balancing of water content in the CNS [5], maintaining ion concentration in extracellular spaces [6,7], enhancing neurotransmitters functionality by promoting their synthesis and uptake [8], and regulation of synaptic transmission and plasticity [7].

The motivation for this study was our interest in the structural support to the CNS by a limited number of (surviving) astrocytes in an injured region of brain or spinal cord through sustainable proliferation as well as the impact of dopamine. The effects of this important neurotransmitter, and its polymerized form, polydopamine (PDA), were studied here in the presence of proliferating astrocytes and naturally occurring oxidation in a primary cell culture environment. Dopamine (DA) has a unique role in motivational behavior in human beings that includes learning, mood swings, and attention. The precursor of DA, l-3,4-dihydroxyphenylalanine (L-DOPA), is widely used in the treatment of Parkinson’s disease [9]. Hyperactivity and hypoactivity of dopamine pathways play a role in schizophrenia [10] and attention deficit hyperactivity disorder (ADHD) [11], respectively. Extracellular dopamine is also known to affect astrocytes by promoting their stellation [12] where the concentration dependent stellation of astrocytic processes is due to cyclic adenosine monophosphate (cAMP)-dependent and cAMP-independent DA receptor signaling. Increased DA concentration also increases the sensitivity of astrocytes to morphological changes through cAMP response element-binding protein (CREB) signaling [13].

Polymerization of DA to PDA is a fast irreversible reaction where dopamine goes through oxidation in basic pH around 8.5 [14] to form intermediate dopamine semiquinone (DSQ) and finally, undergoing an intramolecular cycloaddition reaction to form leucodopaminechrome (DAL). Heteroaromatic 5,6-dihydroxynidole is formed by oxidation of DAL which eventually makes the higher order oligomers, commonly called polymerized dopamine (PDA). The reaction pathway is shown in Figure 1 [15]. The self-oxidation of DA to PDA in cell culture media happens in a similar way and was reported by Clement et al. as observed in case of M14 human melanoma cell line [16].

Polymerization of DA can also be obtained at acidic pH by the process of self-polymerization as described by Chen et al. in environments without oxygen [17]. However, typically the sustainable pH for the primary cell culture environments studied here remained basic, and of course occurred in the presence of oxygen (see methods and results section below). These conditions favored the polymerization of DA to PDA. Not much has been studied on the effect of PDA on astrocytes in a brain microenvironment similar to the one recreated in this study.

A colorimetric assay estimation of DA concentration in the primary brain astrocyte cell culture microenvironment was also studied. This was done by a multivariate regression analysis of the images of the cell culture media and the observed shift in their color as the polymer (PDA) particles started populating the system. A unique image analysis technique, based on the hue of red, green and blue intensities obtained from images of media with PDA, was developed to calculate the dopamine concentrations that started the polymerization in the cellular microenvironments under study. This methodology for estimation of dopamine may spare usage of high-end analytical instruments and techniques, and could improve experimental time constraints by the use of image analysis software as described in the results of the work presented here.

## 2. Materials and Methods

### 2.1. Synthesizing Polydopamine at pH 8.5

Dopamine was polymerized to form PDA by preparing 25 µM, 75 µM and 125 µM dopamine (Sigma-Aldrich, Milwaukee, WI, USA) concentrations in 100 mM bicarbonate buffer. One mL of solution for each concentration of dopamine was placed in cell culture suspension plates and left at room temperature for 3 h following the protocol discussed by Sheng et al. in their study [14].

### 2.2. Preparing Astrocyte Culture Media

Astrocyte culture media comprised Basal Medium Eagle (BME, Sigma-Aldrich, Milwaukee, WI, USA) and Ham’s F-12K media with L-glutamine (ATCC, Manassas, VA, USA), 5% horse serum, 5% fetal bovine serum, and penicillin-streptomycin (Sigma-Aldrich, Milwaukee, WI, USA) as previously described by Daniel and DeCoster [18]. Then, the mixture was filtered using a sterile filtration unit (Thermo Fisher Scientific, Waltham, MA, USA).

### 2.3. Treatment of Cultures of Astrocytes with Dopamine (DA)

Rat brain astrocytes were isolated from the cortex of Sprague-Dawley pups as previously described by Wang et al. [19]. To grow the astrocytes, Astrocyte Culture Media was used. After reaching a high density of growth in flasks, cells were lifted using trypsin/ethylenediaminetetraacetic acid (EDTA, Sigma-Aldrich, Milwaukee, WI, USA), from the flasks. To have a pellet, cells were centrifuged and then plated into a 48 well plate coated with poly-lysine (PLL, Sigma-Aldrich, Milwaukee, WI, USA) at an optimal density of 5000 cells per well and grown at 37 °C, 5% CO_2_/95% air in a humidified incubator. After 72 h of growth, the previously described concentrations of dopamine were introduced to cells. Diffquik staining was used to measure the morphological changes in the cell bodies, 4′,6-diamidino-2-phenylindole (DAPI) staining for observing morphological changes in nuclei of cells [18,20] and thiazolyl blue tetrazolium bromide (MTT) assays for measuring cell viability and metabolism [20] were conducted after 48 h of dopamine treatment.

### 2.4. Pretreatment of Culture Wells with DA and Polydopamine (PDA) Formation

Astrocyte culture media was introduced to 48 well plates coated with poly-l-lysine (PLL) and reserved at 37 °C, in a 5% CO_2_ incubator. Different concentrations of dopamine (20× of: 25 µM, 50 µM, 75 µM, 100 µM and 125 µM final concentrations) were introduced to astrocyte culture media after 24 h of incubation, and before any cells were added, to test for PDA formation. After 48 h, 20× of 5000 astrocyte cells/well (final density) were introduced to the plate which was already treated with dopamine. Diffquick staining for understanding the morphological changes in the cell bodies, DAPI staining for observing morphological changes in nuclei of cells [18,20] and MTT assay for checking cell viability and metabolism [20] were conducted after 3 days of introducing cells to the system.

### 2.5. Toxicity Response of Astrocytes to DA and PDA Formation: Cell Viability and Morphology Changes

Cytotoxicity analyses were done 48 h after dopamine addition or 7 days in vitro (DIV) to estimate the toxic effect of the PDA on astrocyte cells. DAPI staining of the nuclei was undertaken for observing morphology changes in nuclei and MTT assay was carried out to determine cell viability and metabolism. Image Pro^®^ Plus analysis software (Version 7, Media Cybernetics, Inc., Rockville, MD, USA), was used to quantify the data for DAPI staining obtained for every field under the microscope.

### 2.6. Red, Green and Blue (RGB) and Grayscale Analysis for Detection of Dopamine Concentration in Brain Microenvironment

To detect the concentrations of DA from color and intensity changes in the cell culture media due to PDA formation, bitmap analysis (red, green, and blue (RGB) and grayscale) were conducted using Image Pro^®^ Plus 7 software. Before terminating the cells-first or dopamine-first plates, the supernatants of each condition were moved to a new 48-well plate to take pictures using a Leica DMI 6000B inverted microscope (Buffalo Grove, IL, USA). Three different regions of each recorded well for individual concentrations of DA (after the formation of PDA), were analyzed for RGB and grayscale intensities to develop a standard for measurement of unknown concentrations of DA in brain microenvironments. The mathematical model we developed towards this goal was based on multivariate regression [21,22].

## 3. Results

### 3.1. Formation of PDA in the Presence of Bicarbonate Buffer at Basic pH

DA was polymerized to PDA at normal room temperature and was observed to form based on concentration in bicarbonate buffer. Figure 2B–D shows images of PDA aggregates from the 25 µM, 75 µM and 125 µM concentrations, respectively, in 100 mM bicarbonate buffer. From the bright field images, the dopamine concentration-based PDA formation was confirmed. The aggregates of PDA settled on the bottom of the plate used for synthesis of PDA, and appear as dark patterns (Figure 2).

In contrast, in the case of PDA formed in astrocyte media, the aggregation of PDA did not occur. Rather, PDA stayed in colloidal form and did not settle down over the course of time (Figure 3). As shown in Figure 3E–H, bright field imaging could be used to maximize the contrast for this PDA formation. In this same system and in these same wells, phase contrast microscopy was used to focus at the bottom surface of the culture plate where the astrocytes grew (Figure 3A–D).

### 3.2. Morphology Changes of Astrocytes during Polymerization of DA to PDA

Polymerization of dopamine in astrocyte media was observed in the presence of growing astrocytes (Figure 3), and this media had an average pH of 8.2 over the experimental. These basic conditions promoted the polymerization of DA to form PDA, and these conditions also altered the morphology of the treated astrocytes, and changed the metabolic activities of the cells. In Figure 3, in the Diffquik stained phase contrast images of astrocytes (panels 3I–L), the changes related to the overall size of the astrocytes and the number of astrocytes is clearly visible. The full range of Diffquik stained astrocytes treated with 0 dopamine (control) and 25–125 µM dopamine, is also shown in Appendix A, respectively (Appendix A). The cell morphology and cell number changes related to DA treatment in the brain microenvironments was quantified, and are shown in Figure 4 and Figure 5.

The average cell area was calculated based on the Diffquik stained astrocytes on randomly selected 15 astrocytes from different wells under study and the data is based on multiple platings of three sets of experiments (*n* = 3). The morphology changes in the astrocytes are observed as shown in Figure 3I–L. By treatment of 25 µM DA, the formed PDA increased the average size (area) of cells by 19-fold (data not shown) and the length of the astrocytes by 6-fold as shown in Figure 4. In the case of treatment by 125 µM DA, the average cell area went up by 16-fold when compared to the untreated control. There was also a significant increase in the size of nuclei of astrocytes as observed from the DAPI analysis results in Figure 5. The full range of DAPI stained astrocytes treated with 0 dopamine (control) and 25–125 µM dopamine, is shown in Appendix A, respectively (Appendix A). Polymerization of lower concentrations of DA to PDA actually increases the size of the astrocytes whereas, in the case of higher concentrations of DA, the stellation of the processes of astrocytes increases which has already been reported by Galloway et al. [12] and Vaarmann et al. [23]. The polymerization of DA to PDA increased in the presence of astrocytes when compared to DA present in astrocyte media only. The absorbance intensity of PDA in the case of 5000 astrocytes per well treated with 25 µM DA was equivalent to the absorbance intensity of PDA formed in the case of astrocyte media alone treated with 75 µM DA as determined using spectrophotometry. Therefore, the polymerization of DA to PDA was promoted in the presence of astrocytes, supporting the involvement of DA, astrocytes and PDA in the same system.

DAPI staining of the astrocyte nuclei was also done to test for significant changes in the nuclear region as well as to measure the numbers of cells present in the culture after formation of PDA (see Figure 5). There was a significant increase in the average nuclear area of astrocytes with low DA and PDA formation as observed in Figure 5A for 25 µM DA and 50 µM DA-treated astrocytes. The average size of the nuclei was increased by 36.3% in presence of PDA formed by 25 µM DA versus the control (no PDA) conditions. In the case of higher amount of polymerization of dopamine (e.g., 125 µM concentration), the average nuclei area stayed almost the same with a slight (4%) decrease compared to the control. However, although there was an increase in overall size of astrocytes with increasing concentrations of DA, the average number of astrocytes in the culture went down significantly by 73.1% and 93.36% in 25 µM DA and 125 µM DA treatments respectively, as shown in Figure 5B.

The metabolic activity (viability) of the astrocytes is shown in Figure 6 where dopamine with different concentrations was introduced to astrocytes which were plated and growing for 7 days. This process resulted in formation of PDA and the cells demonstrated increased metabolic activity in the presence of the polymer in the system. Astrocyte metabolic activity/viability went up by 17.58% in the presence of PDA polymerized from 25 µM DA when compared to the untreated control. The metabolic activity continued to increase up to 100 µM DA, but the average activity went down slightly in the case of 125 µM DA, but it was still 14% more than the control values.

The significant increase in the average astrocyte cell size, cell length, nuclear area, and metabolic activity in the presence of PDA from polymerization of lower concentrations of DA, supports the hypothesis that administration of dopamine to astrocytes could promote their function, and hence be useful in situations where limited numbers of astrocytes are grown in living scaffolds for treatment of brain injuries and neurodegenerative diseases. As is shown in studies by Katiyar et al. [24], elongation of astrocytes may form dense bundles of aligned cellular processes that could help align seeded neurons. The alignment of astrocytes has also been reported to promote neuronal growth in 3-dimensional collagen gels by East et al. [25].

Slight changes in the pH of the cellular environment were observed during these studies. The pH readings obtained before introducing DA and after formation of PDA are shown in Table 1. Thus, the formation of PDA from DA did not hamper the pH stability of the cell growth environment.

### 3.3. Effect of PDA on Astrocyte Microenvironment Prior to Plating of Cells

When astrocytes were introduced to cell culture wells treated with DA and allowed to polymerize to PDA before cell plating, both cell number and metabolic activity were affected. Figure 7 shows the Diffquick stained phase contrast images of astrocytes under different DA concentrations and allowing 72 h to form PDA. The morphological changes related to the toxic effects of PDA on the overall size and number of astrocytes is clearly visible. The phase contrast images in Figure 7A–D show the growing astrocytes in the presence of PDA in the 2-dimensional culture environment. The bright field images in Figure 7E–H show the change in the color of the media with increase of PDA formation which is directly dependent on the DA concentration introduced in the astrocyte media. The growth inhibition effect of introducing astrocytes into microenvironments with these increasing concentrations of DA and already formed PDA, is shown in Figure 8 as quantified by the MTT assay.

The metabolic activity (viability analysis) of the astrocytes is shown in Figure 8 where dopamine in different concentrations was introduced to astrocyte media resulting in formation of PDA, and were kept in the incubator for 2 days. Thereafter, astrocytes were plated and were allowed to grow for 3 days in the presence of PDA which had already formed in the system. It was observed that an increase in concentration of PDA leads to a significant decrease in the viability/metabolism/cell number of the astrocytes, when DA is introduced to the cell culture media prior to plating of the cells. Thus, the PDA formed in the vicinity of astrocytes decreased the metabolic activity of these cells (e.g., 13.6% lesser viability of astrocytes with PDA formed from 25 µM dopamine in astrocyte media than the control where no PDA was present). The presence of PDA in the system decreased the number of cells in the culture. This growth inhibition effect was also observed in Figure 7I–L. In this study, the growth inhibitory effect of PDA became quite clear from the MTT results and the decrease in number of astrocytes from cell culture studies in astrocyte media with PDA before the plating of the cells.

We believe that this is the first time that growth inhibition of PDA on astrocytes has been reported and clearly observed with two different methods. As far as we know, there has been no report from previous research studies on toxic or growth inhibition responses to PDA or dopamine directly on astrocytes. Nishino et al. [26] found that 3-nitroproprionic acid singly or in combination with dopamine contributed to the injury of astrocytes in the striatum of the brain. The effect of PDA directly on brain cells, including astrocytes, has been largely studied from a biomaterial’s perspective, without indications of toxicity [27,28,29].

### 3.4. Estimation of DA Concentration by Novel Analysis of Images from Polymerization of DA to PDA in Astrocye Cell Culture Microenvironments

With the formation of PDA in astrocyte culture media, the color of the media shifts towards darker brown as shown in Figure 9, and the absorbance intensity thus increased. Polymerization of DA to PDA was observed to be directly proportional to the concentration of DA that was introduced to the astrocyte media.

The change in the absorbance intensity in the astrocyte media was recorded through a wavelength scan (800 nm to 360 nm) using a Beckman Coulter DU800 ultraviolet-visible (UV-Vis) spectrophotometer (Brea, CA, USA). The increase in DA concentration led to an increase in the concentration of PDA and, consequently, a shift of the absorbance towards higher values as shown in Figure 10.

#### 3.4.1. Mathematical Analysis of RGB Color Intensity Microscopy Images after Generation of PDA

RGB and grayscale image analysis was undertaken for the two different sets of experiments described in Section 2.3 and Section 2.4. For the rest of this section, we call the experiments in Section 2.3 “cells-first” and those in Section 2.4 “dopamine-first”. The images were recorded using a Leica DM6000B optical microscope for all 3 sets of trials for each experiment and the RGB and grayscale intensities were extracted by Image Pro 7 image analysis software. We then used the intensity values of these images and the known DA concentrations to develop a model that would help us in the future to quantify any unknown concentration of dopamine from PDA-shifted images of astrocyte cultures in vitro or potentially samples from in vivo studies.

By denoting by *C* the concentration of DA and by *I* = [*I_r_*, *I_g_*, *I_b_*] the RGB intensities per image from the polymerization of DA to PDA, in both experiments (cells-first and dopamine first), *I* depends on *C*. However, each experiment was repeated three times (three trials per experiment). Then, *C* can be obtained by solving the problem of minimization of the summation of the least square errors from each trial per experiment. A short summary of the methodology is given below with the details and proofs in the Appendix A).

We consider the multiple linear regression model *C* = *A × f*(*I*) [30] with *C* being the vector with *n* known distinct concentrations (here *n* = 6), *A* the matrix with the model’s coefficients (to be estimated), and *f*(*I*) as in Equation (2) below, defining ${\bar{I}}_{i}$ as the matrix with elements of the natural logarithm of each color’s intensity for *ith* trial (here *i* = 1, …, 3), that is:

(1)$${\bar{I}}_{i}=\left[\begin{array}{c} \ln(I_{r,1})\ldots \ln(I_{r,n})\\ \ln(I_{g,1})\ldots \ln(I_{g,n})\\ \ln(I_{b,1})\ldots \ln(I_{b,n}) \end{array}\right]$$

As we show in Appendix A, the optimal coefficients *A* that minimize the summation of the least squared errors are estimated by solving the least squares problem *C* = *A × f*(*I*) with *f*(*I*) given by:
(2)$$f(I)={\left(\sum_{i=1}^{3}{\bar{I}}_{i}{\bar{I}}_{i}^{T}\right)\left(\sum_{i=1}^{3}{\bar{I}}_{i}^{T}\right)}^{+}$$
where the superscript + represents the Moore-Penrose pseudo-inverse of a matrix [31]. Having estimated the optimal model’s coefficients *a_r_*, *a_g_*, and *a_b_*, we can then estimate *C* given by the experimentally measured densities *I*, that is:
(3)$$\hat{C}=a_{r}\times \ln(I_{r})+a_{g}\times \ln(I_{g})+a_{b}\times \ln(I_{b})$$
where $\hat{C}$ is the estimated dopamine concentration for any RGB input intensities: *I_r_*, *I_g_*, and *I_b_*. The model coefficients *a_r_*, *a_g_*, and *a_b_* for both experiments (cells-first and dopamine-first) are provided in Table 2.

Inversely, being provided the concentration *C* of DA and trying to find the corresponding RGB intensity values, we basically solve a least squares of errors problem for each of the three colors summing the intensities of each color across trials (see Supplement for proof). Then, if we denote:
(4)$$f(I_{j})=\sum_{i=1}^{3}I_{j}^{\left(i\right)}$$
where the index *i* = 1,…,3 stands for the number of trials, the index *j* for R, G, or B, its exponential relation with *C* in the most general form is given by *f*(*I*_*j*_) = α*_j_* × exp(−β*_j_**C*) with *j* ∈ {*R, G, B*}. Then, solving this exponential model for RGB intensities, we find:
(5)$${\hat{I}}_{r}=\;\alpha_{r}\;\times \;\exp(-\beta_{r}C),\;{\hat{I}}_{g}=\;\alpha_{g}\;\times \;\exp(-\beta_{g}C),\;{\hat{I}}_{b}=\;\alpha_{b}\;\times \;\exp(-\beta_{b}C)$$
where $\hat{I}r$, $\hat{I}g$, and $\hat{I}b$, are now the estimated RGB intensities for any input concentration *C* of dopamine. The coefficients of (5), estimated from our experimental data, are given in Table 3, and validation results for the RGB color intensities at known DA concentrations are illustrated in Figure 11, and results for a wide range of DA concentrations in Figure 12. The average R square is very close to 1, denoting an excellent fit of the model to the data.

#### 3.4.2. Mathematical Analysis of Grayscale Intensity Images from PDA Generated from Dopamine

To estimate the concentration *C* of DA from grayscale intensity *I*_gray_ images requires a simpler model than the one for the RGB color analysis in Section 3.4.1, Equation (3). For grayscale, there is only one regression (independent) variable per experiment and the least squares fit of the (exponential) model with the experimental data generated the following equations for estimation of *I*_gray_ from *C* (as well as for the estimation of *C* from *I*_gray_ by a simple inversion):(6)$$\mathrm{Dopamine}-\mathrm{first}:\;{\hat{I}}_{gray}=\exp(-0.0082C)\;\mathrm{Cells}-\mathrm{first}:\;{\hat{I}}_{gray}=0.9911\exp(-0.0096C)$$

The average R-squared values were close to 1: (0.983 and 0.989 for the dopamine-first, and the cells-first, respectively), denoting a very good fit of the model here too. Projected model values of PDA grayscale intensities for sequential *C* values of DA are given in Figure 13. Comparing the results in Figure 13 with those in Figure 12, we observe that the projected grayscale intensity values appear to be more accurate than those for RGB, over a wide range of dopamine concentrations. This may be due to grayscale intensities having less variance than those for RGB. Furthermore, the input for grayscale images is based only on one independent variable, namely intensity, whereas for RGB images it is based upon three independent variables, namely the individual R, G, and B color values.

## 4. Conclusions and Future Work

In this study we investigated astrocyte proliferation and growth in the presence of dopamine and polydopamine. Metabolically active astrocytes can be introduced in live scaffolds for treatment of brain injury and neurodegenerative diseases [24]. As we have shown in this study, the success of such an endeavor may depend, for example, on the concentration of local neurotransmitters and polymers, in this case dopamine and polydopamine (PDA). Dopamine is a neurotransmitter present in human brain in concentrations ranging from nanomolar to micromolar [32,33]. Through polymerization to PDA, an increased concentration of DA can promote the growth and metabolism of astrocytes as we have shown here, by increasing the cationic elemental concentration in brain microenvironments and by promoting the regenerative process faster than for controls (no PDA). Increase in the concentration of PDA in the cellular environment also adds up to concentration of cationic elements since PDA is positively charged because of its amine group. The presence of cationic polymers enhances the metabolism of cells by promoting their attachment to the culture surface and also in isolating the cells from each other during their proliferation [34]. Therefore, the astrocytes in the presence of PDA demonstrated an overall increase in the total cell size and nuclear area accompanied by high metabolic activity when introduced in a culture where polymerization of DA to PDA was happening. We observed that 25 µM DA in the presence of astrocytes polymerizes to PDA to the same extent as 75 µM DA in astrocyte media without cells. This is confirmation for the direct involvement of astrocytes in the polymerization process. The potential growth inhibitory effects of PDA towards astrocytes had been an unexplored area which we covered in this study and may help shape the usage of PDA coated surfaces in cell culture studies involving structural cells like astrocytes.

At least under the conditions used in our experiments, timing was key for differential effects of dopamine and PDA on astrocytes. We found that for growing cultures of astrocytes (“cells first”), where dopamine was added later, even though morphological changes were observed, the metabolism as measured by the MTT assay was not inhibited (Figure 6). In contrast, astrocytes introduced into culture wells where PDA had already formed (“dopamine first”), again showed morphological changes, but now also showed significantly diminished cell metabolism (Figure 8). Therefore, desired morphological as well as metabolic effects of dopamine and PDA on astrocytes should be closely tuned from a biomaterials standpoint.

Quantification of concentrations of unknown chemicals by a combination of colorimetric assays and image analysis techniques is a novel approach that has the potential to limit the usage of high-end expensive analytical chemistry instrumented techniques such as spectrophotometry, high-performance liquid chromatography (HPLC), and the instrumentation for fluorescence detection. The development of a multivariate linear regression modeling methodology for the estimation of unknown dopamine concentration from in vitro colorimetric images could be useful in the quantification of other unknown chemical concentrations in various colorimetric assays. Examples could include Griess reagent quantification of nitric oxide evolved through catalysis of S-nitosothiols and without even involving UV-Vis readings [35]. This novel approach for quantification of unknown chemical concentrations in various colorimetric assay protocols such as determining chemical oxygen demand through colorimetry [36,37], can significantly lower the cost of analysis. Comparison of results from processing of digital images versus spectrophotometric measurements has been previously carried out [38] recently employing even the use of smartphones [39]. Here we used both spectrophotometric methods (Figure 10) and microscopic methods with digital image analysis to correlate dopamine concentrations to the colored poly-dopamine that was formed. The accuracy of this technique may be improved by increasing the experimental sampling frequency and may be assessed in future studies. Finally, application of the technique to data from in vivo samples, imaged using color or grayscale (monochrome) cameras, may be possible too.

## Figures and Tables

**Figure 1 polymers-12-02483-f001:**
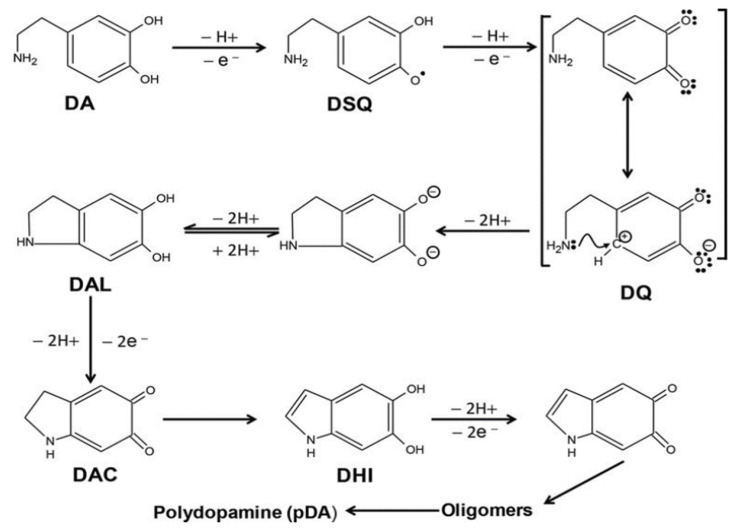
Steps involving autooxidation of dopamine to polydopamine in alkaline pH [15].

**Figure 2 polymers-12-02483-f002:**
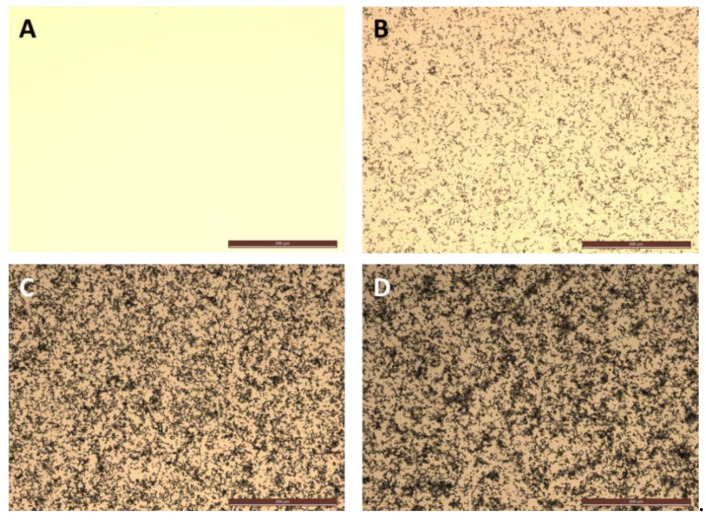
Formation of polydopamine (PDA) after five hours of introducing dopamine (DA) to 100 mM sodium bicarbonate buffer at pH 8.5. (**A**) Sodium bicarbonate buffer; (**B**) 25 µM dopamine in the buffer; (**C**) 75 µM dopamine in buffer; and (**D**) 125 µM dopamine in the buffer. Scale bar indicates 100 µm in all panels.

**Figure 3 polymers-12-02483-f003:**
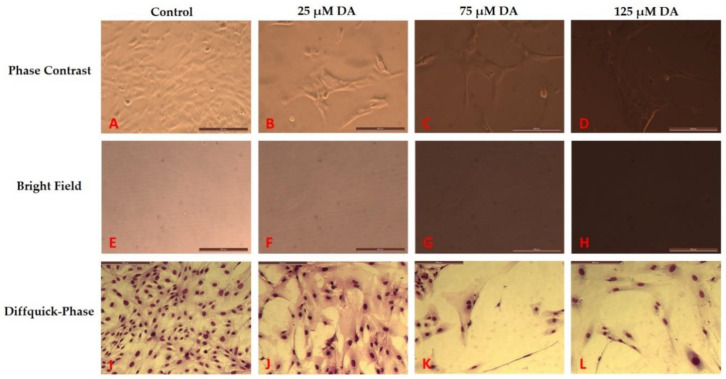
Treatment of 5000 astrocytes/well with dopamine which led to polymerization and formation of polydopamine. Scale bars indicate 200 µm. Panels (**A**–**D**) show phase contrast images of astrocytes (untreated and treated with dopamine, 48 h post treatment). Panels (**E**–**H**) show the bright field images of the same fields mentioned in panels (**A**–**D**). Panels (**I**–**L**) show the Diffquick stained astrocytes (untreated control and treated with dopamine), 48 h after treatment.

**Figure 4 polymers-12-02483-f004:**
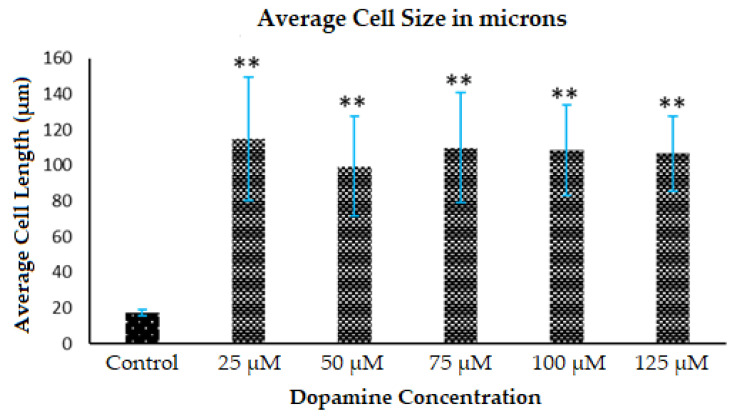
Morphology changes in the astrocyte cell body. Diffquik staining of the astrocytes was carried out to quantify the average cell size by 15 randomly selected astrocytes from randomly selected 3 fields in 2 wells of cultured cells respectively from 3 sets of different experiments treated with the indicated concentrations of DA which in turn formed PDA and the average of their summations were taken. Data represent multiple plating for each condition and ** denotes *p* < 0.01 compared to control condition for cell length analyzed.

**Figure 5 polymers-12-02483-f005:**
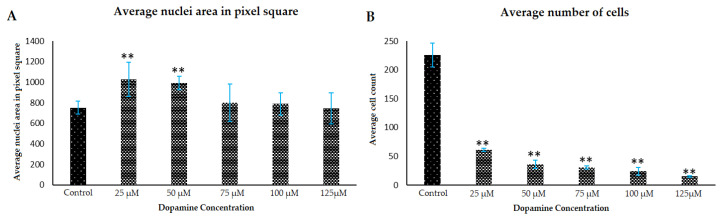
Morphology and number of the astrocyte nuclei in presence of DA and PDA. Astrocytes were plated at 5000 cells/well each, allowed to grow for 5 days, and then were treated with different concentrations of DA, allowing PDA to form for the next 2 days. Thereafter, cells were washed and fixed. DAPI staining (see Methods) was performed to check for morphological changes in nuclei area (panel (**A**)) and number of nuclei (panel (**B**)). Data are averages from multiple platings of cells for each condition and ** denotes *p* < 0.01 compared to control condition.

**Figure 6 polymers-12-02483-f006:**
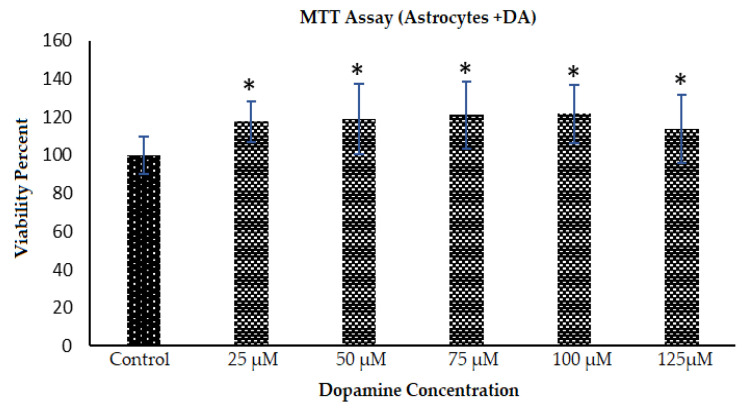
Metabolic activity of incubated astrocytes with DA treatment. Astrocytes were plated at 5000 cells/well each and then allowed to grow for 7 days, and treated with DA at indicated concentrations. PDA was allowed to form for the next 2 days when the MTT viability assay (see Methods) was used to assess the cellular metabolic activity. Data are from multiple platings of cells for each condition and * denotes *p* < 0.05 compared to control condition.

**Figure 7 polymers-12-02483-f007:**
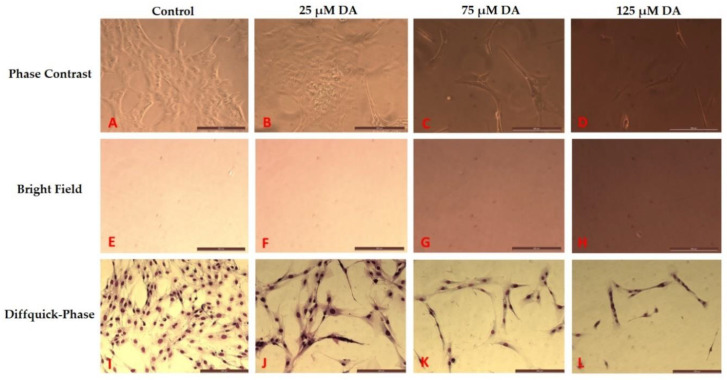
Morphological changes in astrocytes related to formation of PDA prior to cell plating. Astrocytes at a density of 5000 per well, were introduced to wells which had already been treated with dopamine and ongoing polymerization of DA to PDA at the indicated concentrations. Scale bars indicate 200 µm. Panels (**A**–**D**) show phase contrast images of astrocytes (untreated and treated with dopamine, 72 h post plating). Panels (**E**–**H**) show bright field images of the same fields as in panels (**A**–**D**). Panels (**I**–**L**) show Diffquik stained astrocytes (untreated control and treated with dopamine) 72 h post plating.

**Figure 8 polymers-12-02483-f008:**
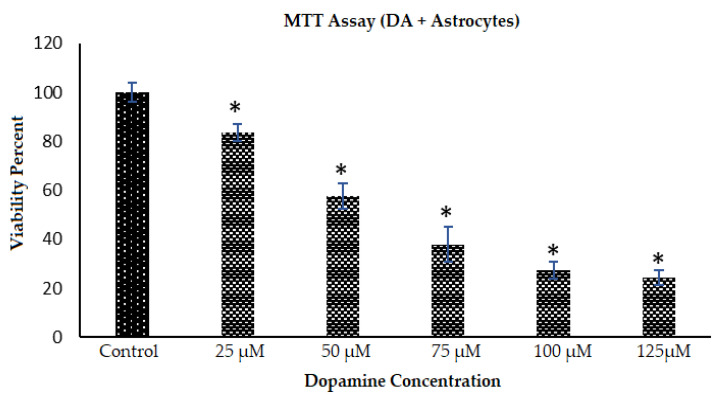
Metabolic activity/viability analysis of astrocytes in the presence of PDA per different DA concentrations. DA was introduced in astrocyte media in the above indicated concentrations. After 48 h in the incubator, astrocytes were plated at 5000 cells/well and allowed to grow for 3 days in the already formed PDA environment. A MTT assay was carried out to measure the cellular metabolic activity in the presence of PDA. Data come from multiple platings of each of the six conditions (bars in the figure), and * denotes *p* < 0.05 compared to the control condition.

**Figure 9 polymers-12-02483-f009:**
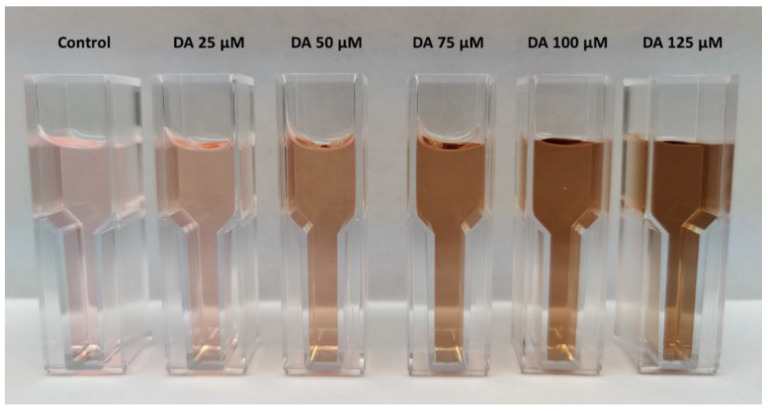
Shift in the color of astrocyte media with self-oxidation of DA to PDA.

**Figure 10 polymers-12-02483-f010:**
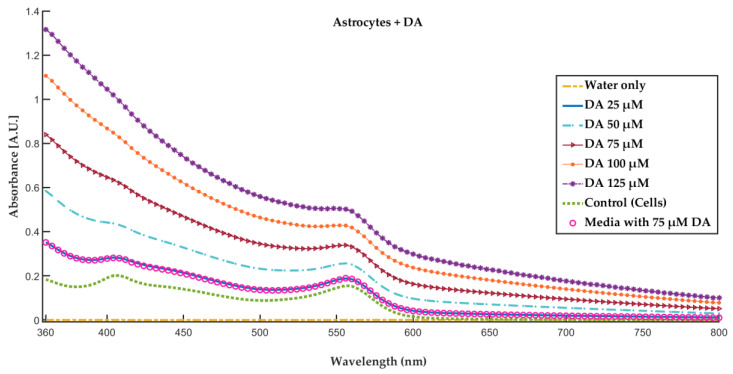
Ultraviolet-visible (UV-Vis) spectrum of water, control, and astrocyte media with different DA concentrations in the presence of cells (astrocytes). The results show the increase in the absorbance intensity with the increase in the polymerization of DA to PDA. Control = cells without added DA. DA concentrations indicate absorbance measured in the presence of cells, except for media alone with 75 µM DA (open circles), which was measured under conditions without cells present.

**Figure 11 polymers-12-02483-f011:**
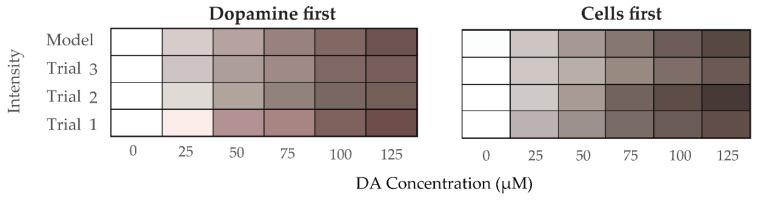
Visual comparison of RGB color intensities acquired experimentally and estimated by the model in (5) as per the trial and experiment.

**Figure 12 polymers-12-02483-f012:**
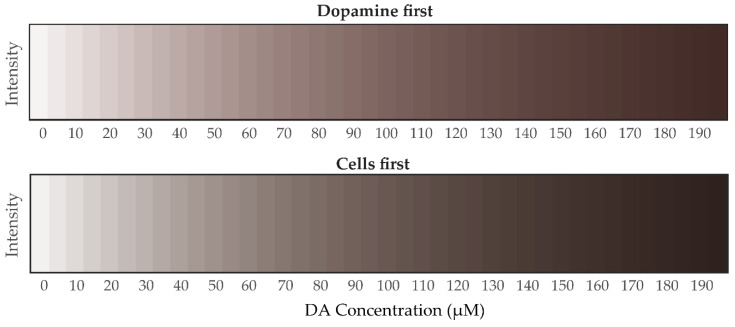
Projected RGB color intensities from the exponential model (5) for a wide range of dopamine concentrations (incremented by 5 µM) for both experiments: dopamine first (**top** panel) and cells first (**bottom** panel).

**Figure 13 polymers-12-02483-f013:**
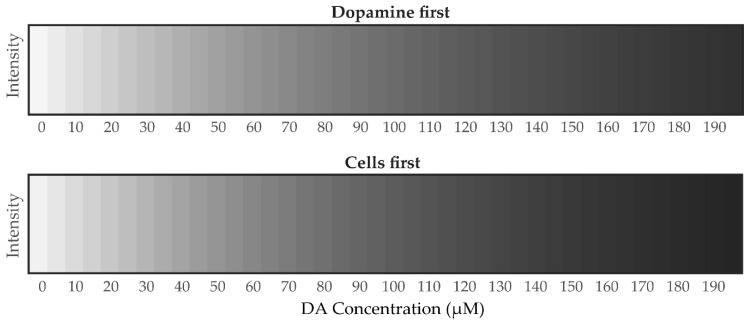
Projected grayscale intensities from the exponential model (6) for a wide range of dopamine concentrations (incremented by 5 µM) for both experiments: dopamine first (**top** panel) and cells first (**bottom** panel).

**Table 1 polymers-12-02483-t001:** pH readings in astrocyte cultures before and after administration of dopamine.

Astrocytes + Dopamine						
	Control	25 µM DA	50 µM DA	75 µM DA	100 µM DA	125 µM DA
Before Treatment with DA (72 h after plating)	8.1	8.1	8.1	8.1	8.1	8.1
After Treatment with DA (48 h post treatment)	8.0	8.15	8.13	8.14	8.2	8.1

**Table 2 polymers-12-02483-t002:** The values of the coefficients in (3) and the coefficient of determination (R^2^) for each trial.

	*a_r_*	*a_g_*	*a_b_*	R^2^		
				Trial 1	Trial 2	Trial 3
Dopamine-first	122.296	292.774	−474.58	0.99	0.984	0.99
Cells-first	354.03	286.25	−640.34	0.995	0.982	0.995

**Table 3 polymers-12-02483-t003:** The values of the coefficients in (5) and the mean of the coefficient of determination (R^2^) of each red, green and blue (RGB) intensity for each trial.

	*α*			*β*			Mean(R^2^)		
	r	g	b	r	g	b	Trial 1	Trial 2	Trial 3
Dopamine-first	1	1	1	0.0068	0.009	0.0083	0.951	0.988	0.99
Cells-first	0.9899	0.9944	0.9934	0.0084	0.0101	0.0107	0.979	0.993	0.993

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
