# Peer review of "Morphological Changes in Astrocytes by Self-Oxidation of Dopamine to Polydopamine and Quantification of Dopamine through Multivariate Regression Analysis of Polydopamine Images"

_polymers, 2020, doi:10.3390/polym12112483_

Round 1

Reviewer 1 Report

In this study, the authors suggested a novel approach analyzing the concentrations of unknown chemicals by combination of colorimetric assays and image analysis techniques. Especially, this new methodology could be alternated in place of other expensive instrumental analytical tools such as HPLC, spectrophotometry, and instrumentation for fluorescence detection. Particularly, new method could calculate the concentration of DA quantitatively. Moreover, it was really interesting to see the results wherein poly-dopamine affects astrocytes morphology, cell numbers, and metabolism.

However, I have two questions described as below.

Major comments

(1) The Fig 6 showed that metabolic activity (i.e., toxicity and viability) of the astrocytes treated with different concentrations of dopamine. This result exhibited that not only morphology of astrocytes but it’s metabolic activity/viability were changed. In contrast, in Fig 8 showing opposite result even though morphology of cell was changed but the metabolic activity and viability percent significantly declined due to cytotoxicity of PDA. Herein, I really wonder about PDA’s toxicity. The authors guessed that a significant decrease in the viability/metabolism/cell number of the astrocytes result from PDA. Ku et al., demonstrated that mammalian cells, such as fibroblasts, osteoblasts, neurons and endothelial cells, cultured on poly-dopamine surfaces showed normal proliferation without cytotoxicity. Based on this reference, I do not agree the authors claim in which PDA leads to significant decrease those factors of the astrocytes due to its toxicity.

(Ku SH, Ryu J, Hong SK, Lee H, Park CB. General functionalization route for cell adhesion on non-wetting surfaces. Biomaterials 31, 2535–2541 (2010))

(Ku SH, Park CB. Human endothelial cell growth on mussel-inspired nanofiber scaffold for vascular tissue engineering. Biomaterials 31, 9431–9437 (2010).)

(2) In the abstract, the authors claimed that this new approach could be used in place of high-end and expensive analytical chemistry instruments, such as spectrophotometry, mass spectrometry, and fluorescence techniques, for quantification of the concentration of DA after polymerization to PDA under in vitro and potentially in vivo conditions. In this regard, the authors should have compared this novel methodology with conventional instrumental analytical tools to appeal their claim. The results of comparison in terms of how much accurately and quantitatively the concentration of unknown samples can be measured and whether they are efficient should be shown.

Author Response

For Reviewer 1, who had 2 comments for revisions, we address them as follows:

Point 1.  “Herein, I really wonder about PDA’s toxicity… Ku et al. demonstrated normal proliferation without cytotoxicity. Based on this reference, I do not agree the authors claim in which PDA leads to significant decrease those factors of the astrocytes due to its toxicity.”

     Response:  We thank the reviewer for these comments and agree that PDA in our studies may have altered cell metabolism and morphology, rather than having a direct toxic effect.  We will therefore remove the term toxicity, and include the references suggested by the reviewer.  Since the references provided by the reviewer looked at PDA on non-wetting surfaces while we used poly-lysine coated cell culture surfaces, our control cell populations were very active and PDA may then have interfered with these substrates.  In contrast, as reference by Ku et al. (suggested by reviewer), PDA may improve cell interactions on non-wetting surfaces).

Point 2.  “the authors should have compared this novel methodology with conventional instrumental analytical tools to appeal their claim.  The results of comparison … should be shown.”

     Response:  We thank the reviewer for this important point, and we have therefore included two new references which compare quantitative analysis comparing digital imaging with spectrophotometric methods, which we in fact also carried out when considering our presented figure #10, which used spectrophotometer instrumentation.  However, we failed to discuss this linkage, and will therefore make the linkage in our discussion section as well as to add these two supportive references:

  1. Morosanova, M.A., Bashkatova, A.S. and Morosanova, E.I., 2019. Spectrophotometric and Smartphone-Assisted Determination of Phenolic Compounds Using Crude Eggplant Extract. Molecules24(23), p.4407.
  2. HAIFA, B.A., BACÂREA, V., IACOB, O., CĂLINICI, T. and ŞCHIOPU, A., 2011. Comparison between Digital Image Processing and Spectrophotometric Measurements Methods. Application in Electrophoresis Interpretation. Applied Medical Informatics.28(1), pp.29-36.

Reviewer 2 Report

The paper by Karan et al. is focused on the study of dopamine polymerization effect on the morphological changes of astrocytes.  The work fits well to the scope of the Journal. The influence of dopamine polymerization on the astrocytes morphology is important both from the basic studies and practical applications. It is an interesting and good paper. Experiments are properly planed. Obtained data are well discussed. I recommend the manuscript for publication in the present form.  

Author Response

We are pleased that Reviewer 2 recommended acceptance and publication of our manuscript as is.